# MMS22L-TONSL functions in sister chromatid cohesion in a pathway parallel to DSCC1-RFC

Janne JM van Schie[1,2] , Klaas de Lint[1,2] , Govind M Pai[1,2], Martin A Rooimans[1,2], Rob MF Wolthuis[1,2], Job de Lange[1,2]

The leading strand–oriented alternative PCNA clamp loader DSCC1-RFC functions in DNA replication, repair, and sister chromatid cohesion (SCC), but how it facilitates these processes is incompletely understood. Here, we confirm that loss of human DSCC1 results in reduced fork speed, increased DNA damage, and defective SCC. Genome-wide CRISPR screens in DSCC1-KO cells reveal multiple synthetically lethal interactions, enriched for DNA replication and cell cycle regulation. We show that DSCC1-KO cells require POLE3 for survival. Co-depletion of DSCC1 and POLE3, which both interact with the catalytic polymerase ε subunit, additively impair DNA replication, suggesting that these factors contribute to leading-strand DNA replication in parallel ways. An additional hit is MMS22L, which in humans forms a heterodimer with TONSL. Synthetic lethality of DSCC1 and MMS22L-TONSL likely results from detrimental SCC loss. We show that MMS22L-TONSL, like DDX11, functions in a SCC establishment pathway parallel to DSCC1-RFC. Because both DSCC1-RFC and MMS22L facilitate ESCO2 recruitment to replication forks, we suggest that distinct ESCO2 recruitment pathways promote SCC establishment following either cohesin conversion or de novo cohesin loading.

## Introduction

To maintain genomic stability, faithful DNA replication must be followed by coordinated transfer of replicated chromosomes to the future daughter cells. To ensure their correct segregation, sister chromatids are paired by the cohesin complex from the moment of their synthesis until anaphase. The basis for sister chromatid cohesion (SCC) is established during DNA replication by the activity of multiple replisome-associated proteins. DNA replication and cohesion establishment are tightly intertwined processes (van Schie et al, 2021).

Based on studies in yeast, SCC factors associated with the replisome are divided in two cohesion establishment groups (Xu et al, 2007; Borges et al, 2013), which may involve different modes of

action (Srinivasan et al, 2020). One pathway involves Tof1-Csm3 (Timeless-Tipin in human), Ctf4 (AND-1 in human), and Chl1 (DDX11 in human) and converts cohesin already bound to unreplicated chromatin into a form that can topologically entrap replicated sister chromatids (Srinivasan et al, 2020). The other pathway depends on Dcc1-RFC (DSCC1-RFC in human) and requires the de novo loading of cohesin by Scc2 (NIPBL in human) (Srinivasan et al, 2020). To maintain SCC, cohesin needs to be stabilized by the SMC3 acetyltransferase ESCO2 (Eco1 in yeast) (Rolef Ben-Shahar et al, 2008; Unal et al, 2008; Zhang et al, 2008). This induces the binding of Sororin to counteract cohesin release by the cohesin antagonist WAPL (Nishiyama et al, 2010). Strikingly, Eco1/ESCO2 defects are synthetically lethal with depletion of cohesin conversion pathway members DDX11, Timeless, and Tipin but functions genetically with DSCC1-RFC (Borges et al, 2013; Abe et al, 2016; Faramarz et al, 2020; Kawasumi et al, 2021), which is difficult to reconcile with the fact that SMC3 acetylation is thought to be required for stable SCC in both establishment pathways.

ESCO2 has three PCNA interaction domains (Moldovan et al, 2006; Bender et al, 2020), an MCM helicase interaction domain (Ivanov et al, 2018; Minamino et al, 2018), and a motif involved in MMS22L binding (Mms22 in yeast) (Zhang et al, 2017; Sun et al, 2019) that all contribute to ESCO2 chromatin recruitment and efficient SCC establishment. MCM can recruit ESCO2 before initiation of DNA replication and protects ESCO2 from CUL4-DDB1-VPRBP and anaphase-promoting complex–mediated degradation (Minamino et al, 2018). These functions do not seem shared by the PCNA and MMS22L interactions of ESCO2 (Sun et al, 2019; Bender et al, 2020), and thus, the different interactions of ESCO2 at the replisome may exhibit specialized functions which remain to be determined.

The alternative PCNA loader DSCC1-RFC, consisting of DSCC1, CHTF8, CHTF18, and RFC subunits 2–5, plays roles in SCC establishment (Terret et al, 2009; Liu et al, 2020; Kawasumi et al, 2021) and error-free DNA replication (Terret et al, 2009; Lopez-Serra et al, 2013; Kawasumi et al, 2021). DSCC1-RFC interacts with POLE, the catalytic subunit of the leading-strand polymerase ε (Polε) (Murakami et al, 2010; Garcia-Rodriguez et al, 2015; Grabarczyk et al, 2018; Stokes et al, 2020). Although this interaction seems mostly dispensable for cohesion establishment (Liu et al, 2020; Stokes et al, 2020), DSCC1-RFC stimulates POLE processivity in vitro (Fujisawa et al,

[1]Department of Human Genetics, Section Oncogenetics, Amsterdam UMC Location Vrije Universiteit Amsterdam, Amsterdam, Netherlands  [2]Cancer Center Amsterdam, Cancer Biology and Immunology, Amsterdam, Netherlands

Correspondence: j.delange1@amsterdamumc.nl

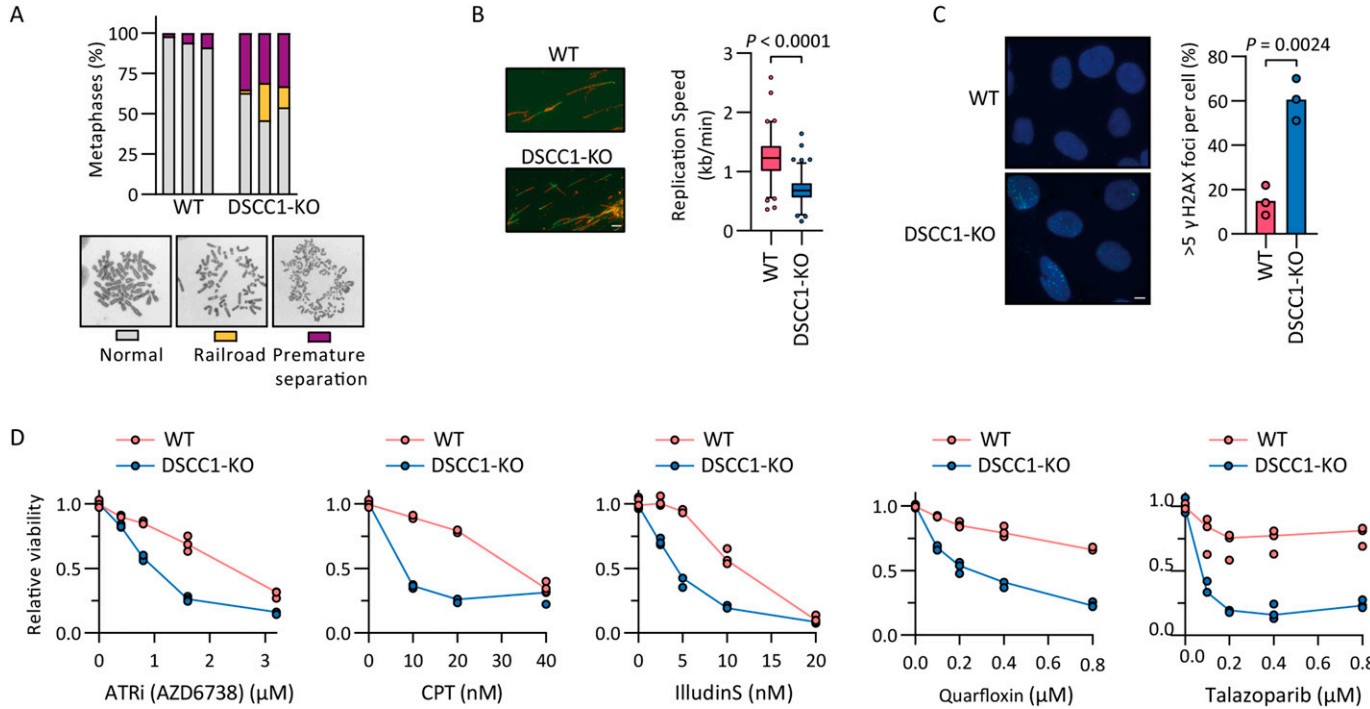

**Figure 1. Human DSCC1-RFC has roles in sister chromatid cohesion, DNA replication, and DNA repair.**
**(A)** SCC defects of indicated cell lines. At least 50 metaphases were scored per condition; three independent experiments are shown as separate bars. **(B)** DNA replication fork speed of indicated cell lines was assessed with a DNA fiber assay. At least 65 fibers were scored per experiment per condition in two independent experiments. Indicated P-value is calculated with an unpaired t test. Scale bar represents 5 μm. **(C)** Cells were assessed for γH2AX foci using immunofluorescence. At least 47 cells were scored per experiment per condition in three independent experiments, and the percentage of cells with more than five foci is shown. The P-value is calculated with an unpaired t test. Scale bar represents 5 μm. **(D)** WT and DSCC1-KO cells were treated with indicated drugs and proliferation relative to untreated cells was assessed after 5 d by a CTB assay. Three technical replicates from a representative of two independent experiments are shown.
Source data are available for this figure.

2017) and loss of the interaction causes DNA replication stress (Garcia-Rodriguez et al, 2015; Stokes et al, 2020). Exactly how DSCC1-RFC contributes to cohesion establishment is under debate. Possibly, DSCC1-RFC promotes Eco1/ESCO2 recruitment by loading PCNA at replication forks (Liu et al, 2020). Alternatively, DSCC1-RFC may facilitate a de novo cohesin loading pathway to enhance chromatin-bound cohesin at the replication fork via an as-yet-unidentified mechanism (Srinivasan et al, 2020; Kawasumi et al, 2021).

To gain further insight in the role of DSCC1-RFC during DNA replication and its role in SCC establishment, we performed genome-wide CRISPR screens in human, isogenic DSCC1-KO and DSCC1-WT cells. These genetic screens yielded synthetically lethal hits involved in cell cycle progression and DNA replication, including several DNA helicases. We selected two identified heterodimers for functional follow-up: the leading-strand Polε POLE3-4 subcomplex and the MMS22L-TONSL heterodimer. We show that loss of POLE3 causes replication defects and in the absence of DSCC1 results in lethal impairment of replication fork progression, further implicating DSCC1 in leading-strand DNA replication. In addition, we find that MMS22L-TONSL, which can bind specific histone marks on replicated DNA, is epistatic with DDX11 and functions in an SCC establishment pathway parallel to DSCC1-RFC. Taken together with earlier reports, we suggest that distinct mechanisms of ESCO2 recruitment may promote SCC

establishment following cohesin conversion or de novo cohesin loading.

## Results

### DSCC1-RFC has roles in SCC, DNA replication, and DNA repair

To investigate the role of DSCC1-RFC in human cells, we created a DSCC1-KO in RPE1-hTERT TP53-KO cells with inducible Cas9 (van der Weegen et al, 2021) (Fig S1A). We confirmed the previously described role of DSCC1-RFC in cohesion establishment (Hanna et al, 2001; Terret et al, 2009; Kawasumi et al, 2021) by quantifying cohesion defects in metaphase spreads (Fig 1A). In addition, in line with previous reports in human (Terret et al, 2009) and avian cells (Kawasumi et al, 2021), DSCC1-KO cells exhibited a reduction in DNA replication fork speed (Fig 1B) and an increase in spontaneous DNA damage, as measured by γ-H2AX staining (Fig 1C). These effects correlate with a modest accumulation in G2 and mitosis (Fig S1B) and a moderate growth rate reduction (Fig S1C). Note that our cellular model is TP53-deficient, which possibly leads to increased resistance to spontaneous DNA damage resulting from DSCC1-KO (Terret et al, 2009). Because genome-wide drug screens have linked DSCC1 loss to sensitization to a large variety of DNA damage–inducing drugs (Olivieri et al, 2020), we tested drug sensitivities in

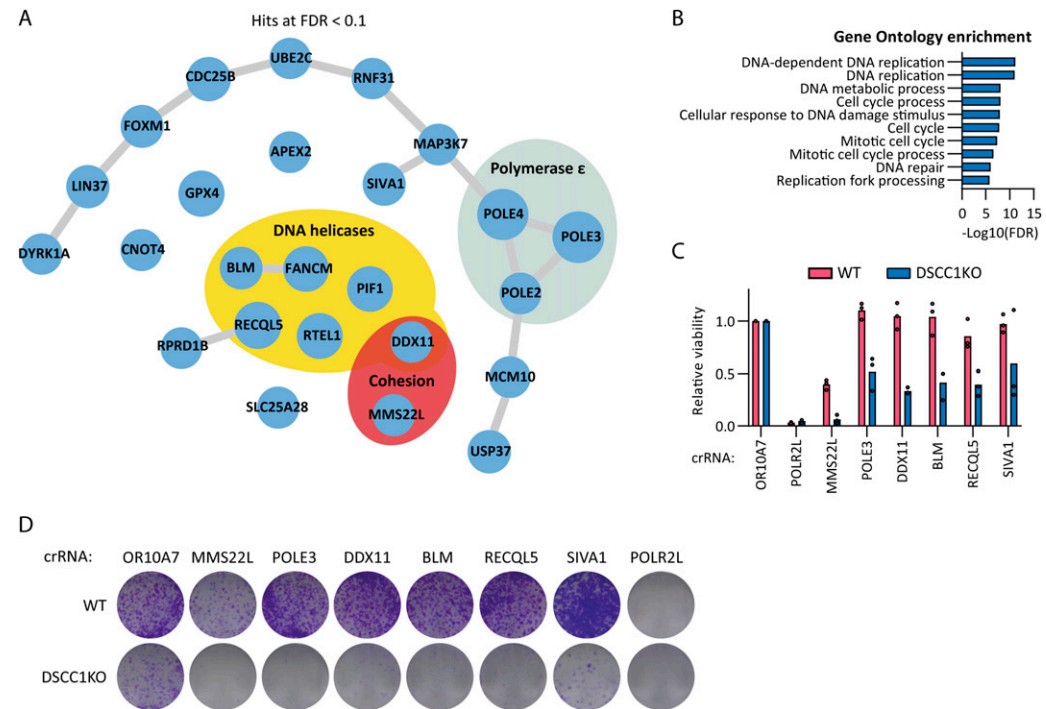

**Figure 2. Genome-wide CRISPR screen reveals multiple synthetic lethal interactions of DSCC1-KO cells.**
**(A)** Network of hits synthetically sick/lethal with DSCC1-KO as determined by DrugZ analysis with an FDR < 0.1. Edges indicate physical protein–protein interactions as annotated in Genemania.org. **(B)** Gene Ontology enrichment of <0.1 FDR hits synthetically sick/lethal with DSCC1-KO extracted from String-db. **(C)** WT and DSCC1-KO cells were transfected with indicated crRNAs and treated with doxycycline to induce Cas9 expression. After 7 d, cell viability was assessed by CTB assay and normalized for cells depleted of the nonessential olfactory receptor OR10A7. POLR2L was included as a common essential control. Dots indicate means calculated from three technical replicates of three independent experiments. **(D)** Clonogenic survival of WT and DSCC1-KO cells 11 d after crRNA transfection and Cas9 induction. Figure shows a representative of two independent experiments.
Source data are available for this figure.

our DSCC1-KO cells. Loss of DSCC1-sensitized cells to the ATR inhibitor AZD6738, the topoisomerase I inhibitor camptothecin, the cytotoxic drug illudin S (which induces transcription- and replication-blocking DNA lesions), the G-quadruplex stabilizer quarfloxin, and the PARP inhibitor talazoparib (Fig 1D). In conclusion, our data validate a role of human DSCC1 in SCC, DNA replication, and the DNA damage response.

### Isogenic CRISPR screens reveal a dependency of DSCC1-KO on DNA helicases, the POLE3-4 heterodimer, and cohesion establishment genes

To examine which genes become essential in the context of DSCC1 loss, we performed a genome-wide drop-out CRISPR screen in DSCC1-KO cells using the TKOv3 library and compared it to the screen we previously performed in the isogenic DSCC1-WT line (van der Weegen et al, 2021). We identified 25 genes whose targeting leads to synthetic sickness or lethality when combined with DSCC1 loss, at an FDR < 0.1 (Fig 2A). The hits were enriched for genes implicated in DNA replication and cell cycle progression (Fig 2B). These included six DNA helicases: BLM, FANCM, RECQL5, PIF1, RTEL1, and DDX11. Previous reports have shown that the iron-sulfur cluster helicase DDX11 is synthetically lethal with DSCC1-RFC in yeast and avian cells, likely resulting from lethal cohesion loss (Xu et al, 2007; Kawasumi et al, 2021). This may relate to the role of DSCC1-RFC in the

de novo cohesin loading pathway, whereas DDX11 appears to be involved in the conversion of preloaded cohesin (Srinisavan et al, 2020). Note that we do not find Timeless, Tipin, and AND-1 (WDHD1) and other genes implicated in the cohesin conversion pathway, possibly because knockout of these genes is lethal in RPE1 (Table S1). The screen also identified three subunits of the leading-strand DNA Polε complex (the accessory subunit POLE2 and the POLE3-4 heterodimer) and MMS22L, a factor linked to SCC in yeast and human cells (Zhang et al, 2017; Sun et al, 2019). To validate the screen results, we transfected crRNA:tracrRNA complexes after Cas9 induction and assessed proliferation by CellTiter Blue (CTB) assays after 7 d (Fig 2C) and clonogenic survival after 11 d (Fig 2D). This confirmed the essentiality of multiple DNA helicases, the PCNA ubiquitin ligase adaptor SIVA1, POLE3, and the cohesion factors DDX11 and MMS22L, specifically in the absence of DSCC1. In conclusion, we identified multiple novel synthetic lethal interactions with DSCC1 loss including genes involved in DNA replication and SCC establishment.

### POLE3 contributes to DNA replication additive with DSCC1-RFC

Intrigued by the prominent synthetic lethal interaction of the accessory Polε heterodimer POLE3-4 with DSCC1-RFC, we further investigated this relationship. POLE3-4 is involved in histone cycling at the replication fork and its loss causes replication stress

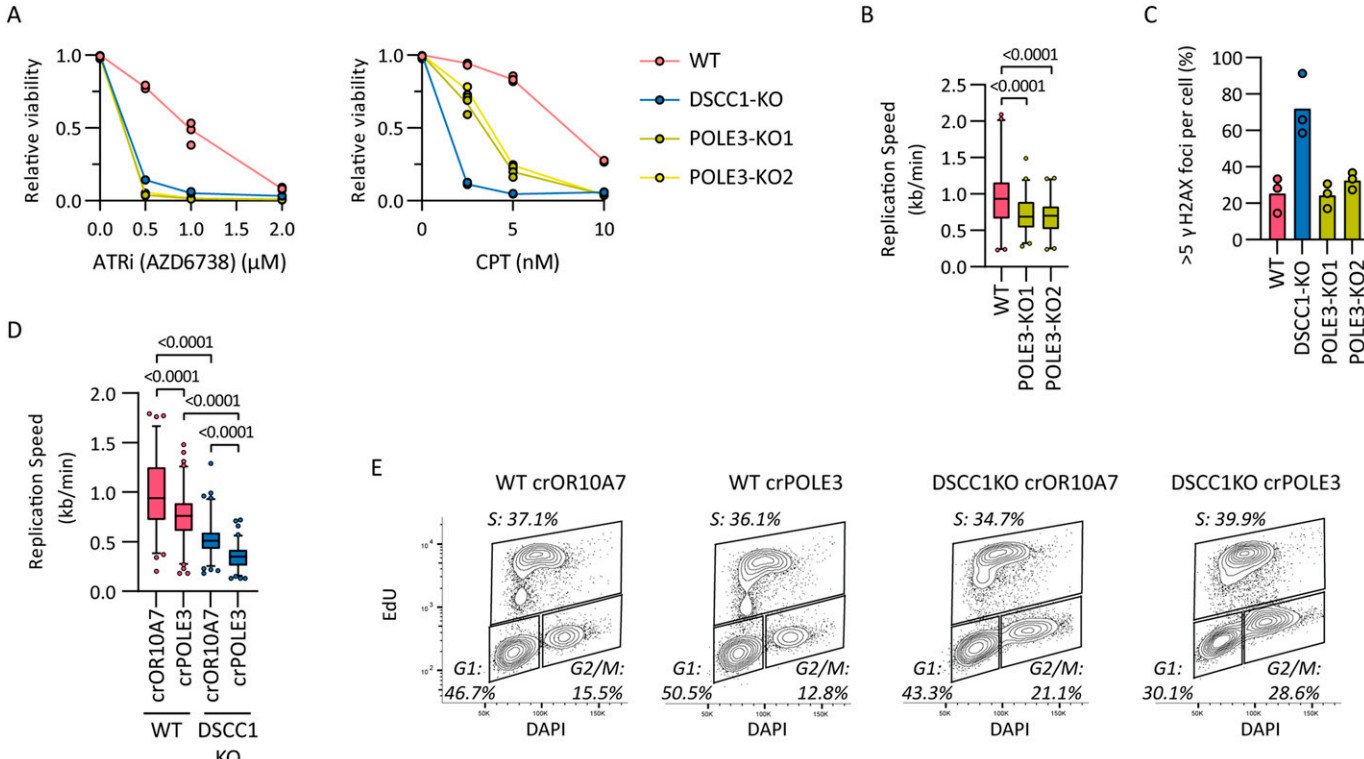

**Figure 3. POLE3 and DSCC1 contribute to DNA replication fork progression in an additive manner.**
**(A)** Relative viability of WT, DSCC1-KO, and two POLE3-KO clones treated with ATR inhibitor (ATRi, AZD6738) or topoisomerase I inhibitor camptothecin assessed by CTB 5 d after addition of drug. Dots represent three technical replicates from a representative of two independent experiments. **(B)** Replication fork speed of POLE3-KOs assessed by a DNA fiber assay. At least 80 fibers were scored per condition. *P*-values were determined by an ordinary one-way ANOVA. **(C)** Cells were assessed for γH2AX foci using immunofluorescence. At least 42 cells were scored per experiment per condition in three independent experiments, and the percentage of cells with more than five foci is shown. Scale bar represents 5 μm. **(D)** WT and DSCC1-KO cells were transfected with indicated crRNAs and treated with doxycycline to induce Cas9 expression. After 5 d, DNA replication fork speed was assessed by a fiber assay. At least 50 fibers were scored per experiment per condition in two independent experiments. Indicated *P*-values are determined by an ordinary one-way ANOVA. **(E)** Flow cytometry analysis of WT and DSCC1-KO cells 5 d after transfection with indicated crRNA:tracrRNAs. Actively replicating cells (S) are identified by EdU incorporation and G1 and G2 by DNA content stained by DAPI.
Source data are available for this figure.

(Bellelli et al, 2018a, 2018b). Both POLE3-4 and DSCC1-RFC interact with the essential, catalytic POLE subunit (Murakami et al, 2010; Bellelli et al, 2018a; Grabarczyk et al, 2018), suggesting both protein complexes function in leading-strand DNA replication. In line with such a functional relationship, POLE3, POLE4, DSCC1, CHTF18, and CHTF8 show a high correlation in co-essentiality score across cell lines as determined by the Cancer Dependency Map database (Fig S2A). To investigate POLE3-4 function, we generated KOs of POLE3 in RPE1-hTERT TP53-KO cells (Fig S2B). POLE3 and DSCC1 both influence sensitivity to ATRi and camptothecin, with DSCC1 playing a more pronounced role in the latter (Fig 3A). As in mouse cells (Bellelli et al, 2018b), POLE3-KO resulted in DNA replication fork progression defects (Fig 3B). However, unlike in DSCC1-KO cells, replication defects in cells that lack POLE3 occurred without an apparent increase in DNA damage signaling (Fig 3C), suggesting both overlapping and distinct functions of DSCC1 and POLE3. In line with parallel functions, depleting POLE3 in DSCC1-KO cells (Fig S2C) exacerbated the delay in fork progression (Fig 3D). Furthermore, POLE3 depletion in DSCC1-KO cells resulted in disturbed nucleotide incorporation in S phase and an increased fraction of G2 cells (Fig 3E), in line with DNA replication problems. Because POLE3-KO

cells have no cohesion defects (Fig S2D) and there is no pronounced increase in cohesion defects or mitotic fractions upon POLE3 depletion (Fig S2E and F), synthetic lethality is unlikely a result of defects in cohesion. Thus, we conclude that the proliferation defects following combined loss of POLE3 and DSCC1 result from detrimental effects on DNA replication.

**The MMS22L-TONSL heterodimer contributes to SCC in parallel with DSCC1**

Another hit from the DSCC1 synthetic lethality screen that caught our attention was MMS22L (methyl methanesulfonate sensitivity protein 22 like; Mms22 in yeast). This protein was previously linked to SCC establishment by recruiting Eco1/ESCO2 to the replisome together with DDB1 (Mms1 in yeast) and CUL4A/B (Rtt101 in yeast) (Zhang et al, 2017; Sun et al, 2019). In vertebrate cells, MMS22L interacts with TONSL—a protein which is apparently lacking in yeast—to form a heterodimer that localizes at replication forks and functions in the DNA damage response (Duro et al, 2010; O'Connell et al, 2010; O'Donnell et al, 2010; Piwko et al, 2016). Consequently, we probed whether the MMS22L-TONSL heterodimer functions in SCC

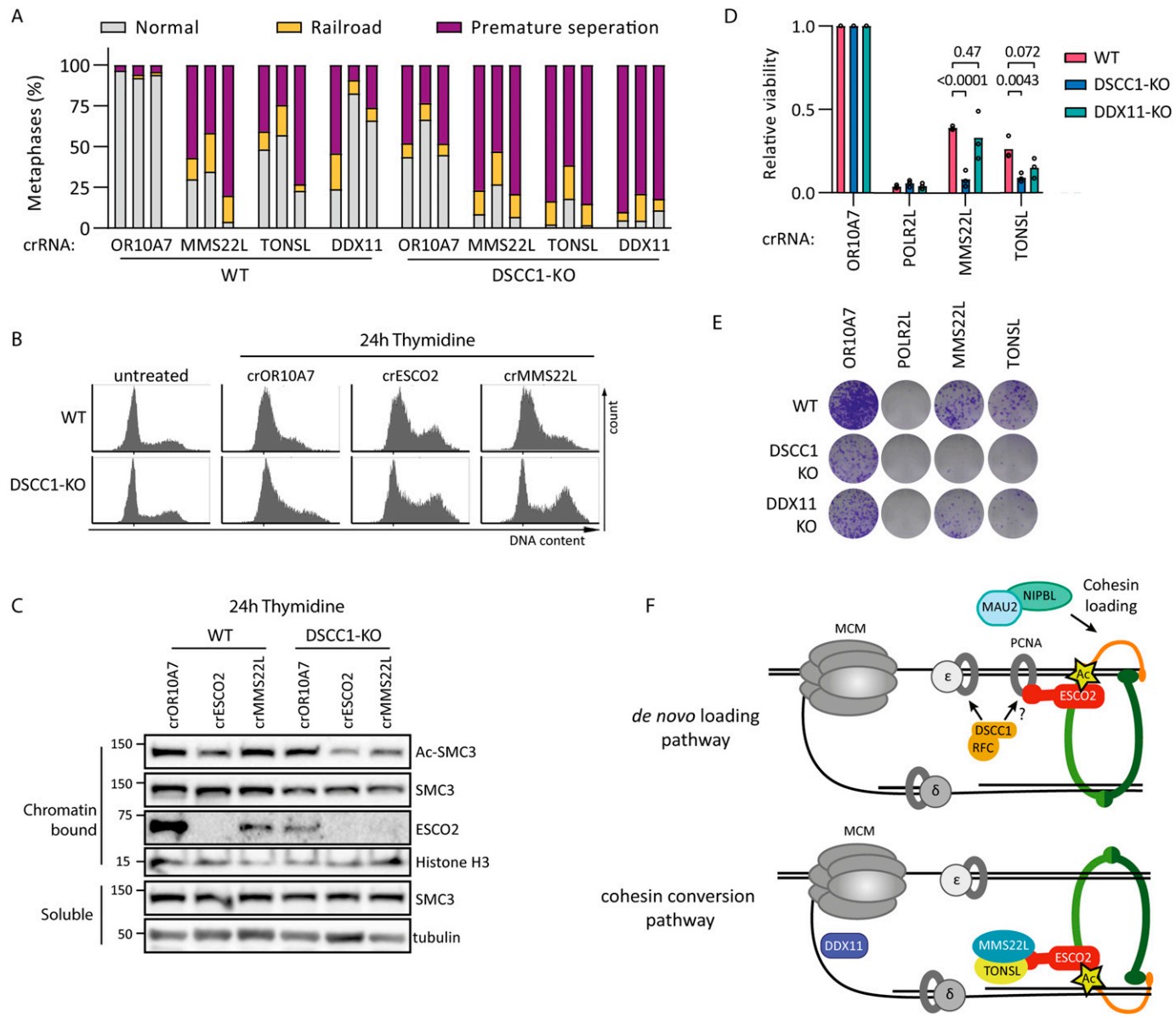

**Figure 4. MMS22L-TONSL heterodimer contributes to sister chromatid cohesion in parallel with DSCC1-RFC.**
**(A)** Cohesion analysis of indicated cell lines 5 d after transfection of indicated crRNA:tracrRNA complexes with induction of Cas9. Each bar represents an analysis from three independent experiments. At least 50 metaphases were analyzed per condition. **(B)** Cells were transfected with indicated crRNAs and treated with doxycycline to induce Cas9. After 2 d, cells were treated with 2 mM thymidine for 24 h and analyzed by flow cytometry. **(B, C)** Western blot of chromatin-bound and soluble protein fractions from cells treated as in (B). **(D)** Relative viability assessed by CTB 7 d after transfection of indicated crRNA:tracrRNA complexes with induction of Cas9. Data points represent means calculated from three technical replicates from three independent experiments. Indicated *P*-values were calculated by an ordinary one-way ANOVA. **(E)** Clonogenic assay 11 d after transfection of indicated crRNA:tracrRNA complexes with induction of Cas9. **(F)** Proposed model of DSCC1-RFC and MMS22L-TONSL functions in SCC establishment pathways. DSCC1-RFC facilitates the de novo cohesin loading pathway, whereas DDX11 and MMS22L-TONSL contribute to the cohesin conversion pathway. ESCO2 can be recruited to chromatin by PCNA, which may be loaded by DSCC1-RFC. MMS22L can also recruit ESCO2 to the replication fork, possibly by binding to newly replicated chromatin. The fact that MMS22L-TONSL and DSCC1-RFC contribute to different SCC pathways may suggest that different ESCO2 recruitment mechanisms preferentially contribute to different SCC pathways.
Source data are available for this figure.

in human cells. We also included DDX11 in our analysis as DSCC1 and DDX11 have been described to function in parallel cohesion establishment pathways in yeast and avian cells (Xu et al, 2007; Kawasumi et al, 2021), and we found DDX11 as a hit in our DSCC1-KO screen in human cells. Because we were unable to generate stable

MMS22L-KO cells, likely because MMS22L is essential (Table S1), we used depletion by crRNA transfection. Depletion of both MMS22L and TONSL resulted in cohesion defects (Fig 4A), suggesting the MMS22L-TONSL heterodimer is involved in cohesion establishment. Furthermore, similar to DDX11 depletion, MMS22L and TONSL

depletion in DSCC1-KO cells led to additional cohesion loss (Fig 4A), suggesting that MMS22L-TONSL and DDX11 function in a cohesion establishment pathway that is parallel to the one involving DSCC1. Building SCC requires the interaction of cohesin with chromatin and the acetylation of SMC3 by ESCO2 during DNA replication. To test whether MMS22L and DSCC1 modulate cohesin on chromatin, we depleted MMS22L in WT and DSCC1-KO cells and arrested cells in S phase by a thymidine block, followed by Western blot of chromatin-bound protein fractions. Note that synchronization of MMS22L- and ESCO2-depleted cells proved difficult (Fig 4B), possibly because of their essential function in cellular proliferation. In line with previous reports (Zhang et al, 2017; Sun et al, 2019; Kawasumi et al, 2021), MMS22L depletion or DSCC1-KO results in a decrease of chromatin-bound ESCO2 and concomitant reduction of acetylated SMC3 (Fig 4C). Combined depletion of MMS22L and DSCC1 resulted in a further reduction, suggesting MMS22L and DSCC1-RFC promote ESCO2 recruitment and cohesin acetylation via parallel mechanisms. Furthermore, although MMS22L and TONSL are synthetically lethal with DSCC1, we found that they are epistatic with DDX11 (Fig 4D and E). Although we cannot exclude that mechanisms other than enhanced cohesion loss contribute to the observed synthetic lethality, together, these observations suggest that the MMS22L-TONSL heterodimer functions in the same SCC pathway as the cohesin conversion factor DDX11 but parallel to DSCC1.

# Discussion

In this study, we generated a network of genetic interactions based on synthetic lethality with loss of human DSCC1. Among the members of this network are the DNA helicases BLM, FANCM, PIF1, and RECQL5. These DNA helicases could be necessary to resolve damaging DNA structures arising from replication problems caused by DSCC1 loss. Alternatively, because DSCC1 loss is known to sensitize cells to a plethora of DNA damaging drugs (Olivieri et al, 2020), increased DNA damage resulting from failed resolution of different helicase substrates may in turn require DSCC1-dependent repair. DSCC1 plays roles in DNA replication and SCC, which we illustrated by functional analysis of a few selected hits from the screen.

We find additive replication defects upon depletion of POLE3 and DSCC1, indicating that they have separate effects on fork progression. DSCC1-RFC has a leading-strand orientation and can interact with POLE, the catalytic subunit of the leading-strand Polε complex (Murakami et al, 2010; Garcia-Rodriguez et al, 2015; Grabarczyk et al, 2018; Stokes et al, 2020). This suggests that DSCC1-loaded PCNA specifically enhances POLE processivity. POLE3-4 is known to stabilize the Polε complex, although loss of POLE3-4 does not completely abolish DNA replication (Bellelli et al, 2018b). When combined, decreased levels of POLE (because of POLE3-4 depletion) and decreased processivity of the remaining POLE (because of DSCC1 depletion) may therefore result in a lethal failure to complete DNA replication. Besides stabilizing Polε, POLE3-4 also exhibits histone cycling activity at DNA replication forks (Bellelli et al, 2018a). Interestingly, POLE3ΔC, which lacks the H3-H4 interaction but retains the interaction with Polε components, displays defective PCNA unloading and RPA accumulation (Bellelli

et al, 2018a). However, POLE3ΔC rescues ATR inhibitor sensitivity (Hustedt et al, 2019), suggesting that the H3-H4 binding is not required for mitigating DNA replication stress. Therefore, it remains to be determined whether the role of POLE3 as H3-H4 chaperone contributes to the observed synthetic lethality with DSCC1.

DSCC1 functions in the de novo loading pathway to establish SCC (Srinivasan et al, 2020; Kawasumi et al, 2021), in line with the synthetic lethality with DDX11 observed in this study. A recent preprint of the Branzei lab shows that PCNA can directly interact with the cohesin loader in yeast and human cells, thus providing a mechanistic explanation for the role of the PCNA loader DSCC1-RFC in SCC (Psakhye et al, 2022 Preprint). In addition, PCNA is known to recruit ESCO2 and thereby promotes SMC3 acetylation (Moldovan et al, 2006; Bender et al, 2020; Liu et al, 2020). Indeed, we find partially impaired ESCO2 recruitment and SMC3 acetylation in DSCC1-KO cells. Interestingly, we also present evidence that MMS22L promotes a separate ESCO2 recruitment pathway: its depletion results in further reduced ESCO2 recruitment, additive cohesion defects, and lethality in DSCC1-KO cells but is epistatic in viability with DDX11. These genetic interactions place MMS22L in the cohesin conversion pathway. Similarly, the function of yeast Mms22 in cohesion is epistatic with the cohesin conversion component Ctf4 (Zhang et al, 2017), in line with a physical interaction of Mms22 and Ctf4 (Gambus et al, 2009; Mimura et al, 2010). Because both PCNA and MMS22L can recruit ESCO2 to the replisome (Zhang et al, 2017; Sun et al, 2019; Bender et al, 2020), we propose that different SCC establishment pathways in part rely on different mechanisms to recruit ESCO2 to the replisome (Fig 4F). It will be interesting to determine if ESCO2, recruited by either PCNA or MMS22L, exhibits different preferences for cohesin arising from different SCC establishment pathways.

In yeast, Mms22 promotes Eco1-dependent Smc3 acetylation and SCC in collaboration with Rtt101-Mms1 (Zhang et al, 2017). This raises the question whether this role is conserved in human DDB1-CUL4. Indeed, DDB1 and CUL4A/B were reported to contribute to SCC in HEK293T cells via MMS22L-dependent ESCO2 recruitment (Sun et al, 2019). However, these genes were not synthetically lethal with DSCC1 in our CRISPR screen. This may be because of poor efficacy of the used guide RNAs, and CUL4A and CUL4B may be partially redundant. Remarkably, DDB1-CUL4 has also been reported to induce ESCO2 degradation by interaction with ESCO2 via VRBP1, which is a substrate recognition component alternative to MMS22L (Minamino et al, 2018). This does not support a major role for CUL4-DDB1 in promoting SCC. Moreover, multiple proteomics studies did not detect an interaction of MMS22L with CUL4A/B or DDB1 but identify TONSL as the main MMS22L interactor in human cells (Duro et al, 2010; O'Connell et al, 2010; O'Donnell et al, 2010). Here, we show that TONSL, which is absent in yeast, contributes to SCC establishment. This suggests potential differences between MMS22L function in yeast and human cells. MMS22L-TONSL can be recruited to chromatin by binding to H4K20me0, a mark of post-replicative chromatin, via a specific domain in TONSL (Saredi et al, 2016). Thereby, MMS22L-TONSL can discriminate replicated from unreplicated DNA (Saredi et al, 2016), theoretically placing it in an ideal position to establish SCC by recruiting ESCO2 exclusively on newly replicated DNA. It will be interesting to assess a potential requirement for DDB1-CUL4 in this process.

# Materials and Methods

### Cell culture

RPE1 cells were cultured in DMEM (Gibco) supplemented with 8–9% FCS, 1 mM sodium pyruvate, penicillin (100 U/ml), and streptomycin (100 μg/ml) at 37°C and 5% $CO_2$.

### CRISPR-Cas9 gene editing

After induction of Cas9 expression by addition of 200 ng/ml doxycycline, RPE1-iCas9 cells were transfected with 20 nM crRNA: tracrRNA duplexes with 1:1,000 Lipofectamine RNAiMAX. To investigate indels, genomic DNA was harvested by incubating cells at 55°C overnight in DirectPCR lysis reagent (Viagen Biotech) with proteinase K. This was used as a template for PCR amplification using the One Taq Hot Start DNA Polymerase kit (New England Biolabs) with appropriate primers (Table S2). Samples were treated with ExoSAP PCR product cleanup reagent (Thermo Fisher Scientific) and analyzed by Sanger sequencing, followed by indel analysis using the online ICE Synthego Analysis tool (Conant et al, 2022).

### Cohesion analysis

To analyze cohesion defects, cells were arrested in metaphase by 20-min treatment with 200 ng/ml colcemid (Sigma-Aldrich). Cells were then harvested and resuspended in 0.075 M KCl for 20 min, followed by fixation in 3:1 methanol:acetic acid. Cells were dropped on microscope slides, and chromosomes were stained by a 4% Giemsa solution. Metaphases with 0–4 railroad chromosomes were classified "normal," cells with more than four railroad chromosomes were classified "railroad," and cells with at least two entirely separated sister chromatid pairs were classified "premature separation."

### DNA fiber analysis

Cells were labeled with 25 μM chlorodeoxyuridine for 20 min, followed by 250 μM iododeoxyuridine for 20 min. Next cells were washed twice with ice-cold PBS and harvested. 2 μl of cell suspension on super-frost microscope glass slides was lysed by addition of 7 μl spreading buffer (200 mM Tris–HCl [pH 7.4], 50 mM EDTA, 0.5% SDS). After 2-min incubation, slides were tilted slightly to allow the drop to run, to spread the DNA fibers over the surface of the microscope slide. After fixation in 3:1 methanol:acetic acid, DNA fibers were denatured by incubation in 2.5 M HCl for 1 h and 15 min. Next, slides were blocked in blocking solution (PBS + 1% BSA + 0.1% Tween-20), followed by incubation with rat anti-BrdU (1:500, Clone BU1/75; Novus Biologicals) and mouse anti-BrdU (1:750, Clone B44; Becton Dickinson) for 1 h. After washing with PBS, antibodies were fixated by 4% paraformaldehyde, followed by incubation with anti-rat Alexa Fluor 555 (1:500; Life Technologies) and anti-mouse Alexa Fluor 488 (1:500; Life Technologies) for 1.5 h. After washing with PBS, slides were mounted and analyzed by fluorescence microscopy. Images were acquired using a fluorescence microscope (Leica), and DNA fiber length was measured in ImageJ. Fork speed was calculated by converting fiber length to kb by using a 1 μm = 2.59 kb conversion factor, followed by division by the chlorodeoxyuridine and iododeoxyuridine labeling time (20 min).

### Cell titer blue and clonogenic assays

After performing indicated treatments, viability was assessed using CTB and/or clonogenic assays. For CTB assays, cells were incubated for 3.5 h with 20 μl CTB reagent (Promega) in 100 μl medium in a 96-well plate in triplicate and was measured at $560_{Ex}/590_{Em}$ with an Infinite F200 microplate reader (Tecan). For clonogenic assays, 1,500 cells were seeded in a six-well plate and after 8 d were fixed in 100% ice-cold methanol followed by staining in 0.5% crystal violet.

### γH2AX foci immunofluorescence assay

Cells on coverslips were fixed with 4% paraformaldehyde, followed by blocking in 3% BSA, 0.3% TX-100 in PBS. Coverslips were incubated with anti-γH2AX (Histone H2A.X phospho-Ser139; 1:500). After washing with PBS, cells were incubated with anti-mouse Alexa Fluor 488 (1:500; Life Technologies). Coverslips were washed with PBS, followed by mounting with ProLong Gold Antifade Mountant with DAPI. Images were acquired using a fluorescence microscope (Leica), and γH2AX foci per cell were quantified using ImageJ.

### Flow cytometry

Cells were labeled with 10 μM 5′-ethynyl-2′-deoxyuridine for 10 min, harvested, and fixed in 4% paraformaldehyde followed by an overnight fixation in 70% EtOH at −20°C. Cells were permeabilized in 0.5% Triton X-100, blocked with 5% FCS, and incubated with Histone H3 pS10 Alexa Fluor 647 (BioLegend) in 1% BSA. Cells were incubated for 30 min with 5′-ethynyl-2′-deoxyuridine Click-iT reaction mixture (Invitrogen), washed, resuspended in 1% BSA with DAPI, and detected by flow cytometry on a BD LSRFORTESSA X-20 (BD Biosciences). Data analysis was performed using FlowJo V10.

### CRISPR-Cas9 screen

The RPE1-iCas9-DSCC1-KO CRIPSR screen was performed in three independent replicates at an average 400-fold sgRNA representation. Briefly, three DSCC1-KO cell populations were transduced in parallel in medium containing 8 μg/ml polybrene with the lentiviral TKOv3 library (a kind gift from K Chan, A Tong, and J Moffat [Donnelly Centre, University of Toronto]) at a multiplicity of infection of ~0.2. After selection with 5 μg/ml puromycin, t = 0 samples were taken and Cas9 was induced by addition of 200 ng/ml doxycycline. Cells were allowed to proliferate for ~12 population doublings for about 2 wk, after which an endpoint sample was taken for each of three populations. Genomic DNA was extracted from t = 0 and endpoint samples using a QIAGEN Blood and Cell Culture DNA Maxi kit. 192 μg of genomic DNA was amplified for each sample using a KAPA HiFi ReadyMix PCR kit and CRISPR_PCR1 primers in 64 parallel 50 μl reactions. In a second PCR reaction, Illumina adapters and i5 and i7 index sequences were added, and PCR products were purified using a QIAquick PCR purification kit. Samples were sequenced on an Illumina NovaSeq6000, and reads were mapped to the TKOv3 library

sequences without allowing mismatches. After normalization of endpoint samples to t = 0 samples, the CRISPR screen was analyzed using DrugZ (v.1.1.0.2) (Colic et al, 2019). For the extended CRISPR-Cas9 screen method, see the sheet "Extended screen method" in Table S1.

### Immunoblotting

Cells were lysed for 10 min on ice in lysis buffer (50 mM Tris–HCl, pH 7.5; 150 mM NaCl; 10% glycerol; 0.2% NP-40; protease and phosphatase inhibitors [Roche]) and centrifuged at 1,300$g$ for 10 min; supernatant contained the soluble protein fraction. The chromatin-bound protein fraction was obtained by washing the pellet twice and incubating for 2 h on ice in lysis buffer with 5 mM $MgCl_2$ and Benzonase (5 U/μl). Proteins were separated by 4–15% or 8–16% Mini-PROTEAN Precast Protein gels (Bio-Rad) and transferred to an Immobilon-P PVDF membrane (Millipore). Membranes were blocked in 5% dry powdered milk in TBST-T and incubated in primary and secondary peroxidase conjugated antibodies. Antibodies used are mouse anti-acetyl-SMC3 (gift from K Shirahige), rabbit anti-SMC3 (Bethyl Laboratories), rabbit anti-Sororin (Santa Cruz), and rabbit anti-Histone H3 (Cell Technology). Protein bands were visualized by chemoluminescence (ECL prime; Amersham).

## Data Availability

All data generated during this study are included in this article and its supplementary information files.

## Supplementary Information

## Acknowledgements

We thank Gerben Vader for critically reading the manuscript. This work was supported by the Dutch Cancer Society (KWF grants 10701 and 13645 to J de Lange).

## Author Contributions

JJM van Schie: conceptualization, investigation, and writing—original draft.
K de Lint: investigation and writing—review and editing.
GM Pai: investigation and writing—review and editing.
MA Rooimans: investigation.
RMF Wolthuis: conceptualization, supervision, and writing—review and editing.
J de Lange: conceptualization, supervision, funding acquisition, investigation, and writing—original draft.

### Conflict of Interest Statement

The authors declare that they have no conflict of interest.

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
