## [Reviewer comments · Life Science Alliance]

Life Science Alliance

MMS22L-TONSL functions in sister chromatid cohesion in a pathway parallel to DSCC1-RFC

Janne van Schie, Klaas de Lint, Govind Pai, Martin Roimans, Rob Wolthuis, and Job de Lange

DOI: <https://doi.org/10.26508/lsa.202201596>

Corresponding author(s): Job de Lange, Amsterdam University Medical Centers

Review Timeline:

Submission Date:	2022-07-11
Editorial Decision:	2022-08-07
Revision Received:	2022-11-04
Editorial Decision:	2022-11-28
Revision Received:	2022-11-30
Accepted:	2022-11-30

Scientific Editor: Novella Guidi

Transaction Report:

August 7, 2022

Re: Life Science Alliance manuscript #LSA-2022-01596

Dr. Job De Lange
Amsterdam University Medical Centers
Human Genetics
de Boelelaan 1118
Amsterdam 1081 HV
Netherlands

Dear Dr. De Lange,

Thank you for submitting your manuscript entitled "MMS22L-TONSL functions in sister chromatid cohesion in a pathway parallel to DSCC1-RFC" to Life Science Alliance. The manuscript was assessed by expert reviewers, whose comments are appended to this letter. We invite you to submit a revised manuscript addressing the Reviewer comments.

Thank you for this interesting contribution to Life Science Alliance. We are looking forward to receiving your revised manuscript.

Sincerely,

B. MANUSCRIPT ORGANIZATION AND FORMATTING:

Reviewer #1 (Comments to the Authors (Required)):

Interested in the role of the DSCC1-RFC complex to DNA replication and sister chromatid cohesion (SCC), the authors generated DSCC1 KO in RPE1-hTERT TP53KO cells and conducted CRSIPR-CAS9 synthetic lethal screens, revealing two important synthetic lethal (SL) interactions. One involves DSCC1 and POLE3, which the authors show to be associated with severe replication defects, the other one involves MMS22L-TONSL. The authors show that MMS22L-TONSL facilitates SCC jointly with DDX11 and in parallel with DSCC1. These results are important for the field and could not be predicted based on the current literature primarily limited to a synthetic lethal interaction between CTF18 (another component of the DSCC1-RFC complex) and DDX11 due to severe cohesion defects and reported roles for budding yeast Mms22 in SCC. However, as such, the paper seems to be a bit dry and does not offer much insight into the process.

I think the paper would benefit by trying to address or at least comment on a few points listed below:

- 1) Is the SL between DSCC1 and POLE3 loss associated with POLE3 role as H3-H4 chaperone?
- 2) Is the discovered function of MMS22L-TONSL shared also by Rtt101-Mms1(DDB1-CUL4) shown to facilitate cohesion in yeast (Zhang et al, EMBO Reports, 2017)?
- 3) Mms1 and Mms22 were shown to stabilize the yeast replisome upon replication stress caused by HU treatment (Vaisica et al, MBC, 2011). Does MMS22L functions jointly or in parallel with POLE3-POLE4 in ensuring replication?
- 4) Although ESCO2 has a PIP in its N-terminus, the recruitment to chromatin in yeast can still happen in the absence of that domain (Moldovan et al, Mol Cell, 2006). Does MMS22-TONSL impair ESCO2 recruitment to chromatin, providing thus an alternative mechanism for ESCO2 recruitment and SCC establishment?

Reviewer #2 (Comments to the Authors (Required)):

Sister chromatid cohesion (Cohesion) is crucial to ensure orderly segregation of chromosomes. Cohesion is established by co-entrapment of sister DNAs within cohesin rings during S-phase. The precise mechanism by which this is achieved is still unclear. A set of conserved proteins (but non-essential in yeast) associated with the replisome are important for cohesion establishment. They are thought to promote cohesion by two independent pathways that operate in parallel during S-phase. Analysis of cohesion of small circular mini-chromosomes in yeast suggests that Chl1 (DDX11), Ctf4 (AND1), Tof1 (TIMELESS) and Csm3 (TIPIN) are important to build cohesion using DNA associated cohesin rings. The Ctf18-RFC (DSCC1-RFC) is important for loading soluble cohesin during S-phase to establish cohesion (Srinivasan et al eLife 2020). In the present study, Janne van Schie et al characterise the DSCC1-RFC in human cells (Ctf18-RFC). They generate DSCC1 knockout cells (in a P53 deficient cell line) and find that this leads to cohesion defects, reduction in replication fork speed and accumulation of -H2AX foci. They then perform a CRISPR screen to look for genes that become essential in the absence of DSCC1 and identify several helicases (including DDX11), subunits of the polymerase epsilon complex and MMS22L. They find that loss of POLE3 in DSCC1-KO cells leads to additive defects specifically in DNA replication, while the loss of MMS21L leads to additive cohesion defects. Based on their observation that DDX11 is epistatic to MMS21L, they suggest that MMS21L and DSCC1 operate in parallel pathways for cohesion.

The observations reported here are interesting, they confirm the previously observed (in yeast and avian cells) genetic interaction between CTF18-RFC and DDX11. The present study also highlights the importance of MMS22L for cohesion in human cells (previously observed in yeast). The experiments are well executed, and the data are clear. The study reaffirms the evolutionary conservation of parallel pathways operating during S phase to establish cohesion and thus warrants publication. I have some comments:

1: The authors measure additive defects in DNA replication in the POLE3 DSCC1-KO double mutant cells, they conclude that the synthetic lethality of the DSCC1 cells upon POLE3 deletion is therefore specific to this additive defect DNA replication (perhaps reflected in the speed of replication). This might well be the case, there are reports in yeast cells that an interaction

between CTF18-RFC and Pol epsilon is not necessary for cohesion (Grabarczyk et al 2018). However, comparing figure 1A with figure S3E it looks like there is a noticeable and perhaps significant increase in the cohesion defects in DSSC1-KO cells upon POLE3 deletion. Could the authors comment on this?

2. Wapl depletion is shown to suppress the synthetic lethality of the CTF18 DDX11 double mutants in avian cells (Kawasumi et al 2021). If the authors wanted to make a strong case that the synthetic lethality of DSSC1 with POLE3 and DDX11/MMS21 was indeed because of specific defects in replication and cohesion respectively. One would expect Wapl depletion to suppress the synthetic lethality of DSSC1 with DDX11/MMS21 and not with POLE3. This is perhaps not within the remit of the current manuscript, in which case, the authors should be a bit cautious about assigning specificities for the synthetic interactions.

3. In Discussion the authors state: Furthermore, since it is conceivable that all cohesin needs to be acetylated in order to contribute to SCC, it is unclear why DSSC1-RFC would specifically contribute to the de novo loading pathway (Srinivasan et al., 2020).

There is a flaw in their logic here. Eco1/ESCO1 abbreviated for establishment of cohesion is a misnomer. It is clear in almost every living system tested thus far that Eco1 is not fundamental for establishment of cohesion in the absence of Wapl mediated release. The analysis of ctf18 mutants in yeast that were defective in Wapl mediated release (and therefore emancipated from Eco1) revealed that they were defective in de novo generation of cohesion in S-phase. The authors observe that levels of Smc3, acetylated Smc3 as well as Sororin on chromatin is lower in DSSC1-KO cells and this is exacerbated by deletion of DDX11 or MMS22L. These observations are consistent with a defect in generation of cohesion in the absence of DSSC1-RFC.

Reviewer #3 (Comments to the Authors (Required)):

The authors investigate DSSC1-RFC, the leading strand alternative PCNA clamp loader. DSSC1-RFC facilitates DNA replication, repair and sister chromatid cohesion (SSC). Here, they perform a genome-wide Crispr screen to find mutated human genes that have negative synthetic interactions with DSSC1-KO. Previous work identified two pathways involving replisome associated proteins in establishment of SSC, one defined by DSSC1 and the other by DDX11. In a model proposed by Nasmyth and coworkers, DSSC1 generates cohesion with cohesin loaded at the replication fork whereas DDX11 generates cohesion with cohesin loaded on unreplicated DNA. As shown here and elsewhere, synthetic lethality occurs when both DSSC1 and DDX11 are mutated. The author's Crispr screen identified many genes involved in DNA replication and cell cycle progression but they follow up with just two. The first is MMS22L, which together with TONSL forms a protein complex at the replication fork. As shown here and in yeast, MMS22L participates in SSC. Since the authors show that MMS22L and DSSC1 contribute to SCC in additive ways, they conclude that they act in different but parallel pathways. They infer that MMS22L-TONSL contributes via the pathway defined by DDX11, which is supported by earlier prior work in yeast. Their work builds on work from Nasmyth and Zou labs by showing that human MMS22L participates in a different cohesion establishment pathway (presumably the DDX11 pathway) than the pathway defined by DSSC1. Putting all this together in humans by an unbiased screen is a timely and worthy advance.

The second gene characterized is POLE3, which together with POLE4, generates a protein complex that, like DSSC1, associates with the leading strand polymerase. They show that POLE3 and DSSC1 contribute to replication in additive ways, suggesting that they act in different but parallel pathways. Only loss of DSSC1 causes H2AX foci, a marker of DNA damage. POLE3 mutation does not create cohesion defects on its own and does not exacerbate defects in DSSC1-KO. Thus, the synthetic relationship between the two is not likely related to cohesion.

Major issues:

1) In figure 4C, the authors claim that there is an additive effect on levels of chromatin-bound cohesin, Smc3 acetylation and sororin when MMS22L and DSSC1 are co-depleted. That effect is imperceptible by eye, particularly when compounded by slight differences in loading (the H3 control), which are noticeable by eye. Without quantitation, I am not convinced. The weakness of this panel maybe reflected by the fact that the authors don't mention it in the Discussion. This should be remedied with quantitation or removed.

2) In figure 4C, the band labelled "Total Smc3" should just say Smc3 since it is only measuring the Smc3 in the chromatin pellet. This experiment should also include a control that either shows that the Smc3 level in whole cell lysate does not change or that the level of Smc3 in the soluble fraction increases.

3) In figure 4A, the authors show that MMS22L and DDX11 depletions both cause additive effects on cohesion when either is combined with DSSC1-KO. This suggests that both operate in pathways parallel to DSSC1. Does this mean they operate in the same parallel pathway? It should be possible to test by co-depletion of MMS22L and DDX11. Otherwise, the authors should reiterate that MMS22L and DDX11 are placed in the same pathway by the yeast data of Zou et al.

3) Discussion - Zou and coworkers showed that MMS22L and its partners (DDB and CUL4) are required for proper cohesion in yeast. Notably, these MMS22L partner proteins were not picked up in the Crispr Screen. This is troubling because the authors argue that synthetic lethality arises from compounding cohesion loss by depletion of both MMS22L and DSSC1. This should either be tested directly by depleting DDB and CUL4 in the DSSC1-KO, or by finding a better explanation. In the second to last paragraph, the argument that MMS22L must have "shared and separate" roles in ESCO2 regulation is irrelevant if cohesion is

the only role that matters for synthetic lethality.

Minor issues:

- 1) Page 4, first paragraph - Reference to figure 3F is wrong.
- 2) Top sentence of figure 6, "but is epistatic with DDX11." - Clarify that epistasis regards viability (fig 4D-E), not cohesion (fig 4A).
- 4) Middle of page 3 - "This may relate to the role of DSCC1-RFC in de novo cohesin loading" should read "de novo cohesin loading pathway" since the authors acknowledge that the mechanistic bases for the two pathways have not been worked out, as they properly state in the last paragraph of the Discussion.
- 5) On page 5 - "cohesion defects caused by DSCC1-RFC loss cannot be rescued by ESCO1 or ESCO2 overexpression (Kawasumi 2021)." In my view this is a tricky citation since it is a negative result, and there are many ways that an overexpression experiment can fail.
- 6) The figure legends are skeletal, thereby forcing the reader to look in other places to find out what was done exactly.

Reviewer #1 (Comments to the Authors (Required)):

Interested in the role of the DSCC1-RFC complex to DNA replication and sister chromatid cohesion (SCC), the authors generated DSCC1 KO in RPE1-hTERT TP53KO cells and conducted CRSIPR-CAS9 synthetic lethal screens, revealing two important synthetic lethal (SL) interactions. One involves DSCC1 and POLE3, which the authors show to be associated with severe replication defects, the other one involves MMS22L-TONSL. The authors show that MMS22L-TONSL facilitates SCC jointly with DDX11 and in parallel with DSCC1. These results are important for the field and could not be predicted based on the current literature primarily limited to a synthetic lethal interaction between CTF18 (another component of the DSCC1-RFC complex) and DDX11 due to severe cohesion defects and reported roles for budding yeast Mms22 in SCC. However, as such, the paper seems to be a bit dry and does not offer much insight into the process.

I think the paper would benefit by trying to address or at least comment on a few points listed below:

1) Is the SL between DSCC1 and POLE3 loss associated with POLE3 role as H3-H4 chaperone?

We address this point in an updated second paragraph of the Discussion:

“We find additive replication defects upon depletion of POLE3 and DSCC1, indicating that they have separate effects on fork progression. DSCC1-RFC has a leading strand orientation and can interact with POLE, the catalytic subunit of the leading strand pole complex (Garcia-Rodriguez et al., 2015; Grabarczyk et al., 2018; Murakami et al., 2010; Stokes et al., 2020). This suggests that DSCC1-loaded PCNA specifically enhances POLE processivity. POLE3-4 is known to stabilize the Pole complex, although loss of POLE3-4 does not completely abolish DNA replication (Bellelli et al., 2018b). When combined, decreased levels of POLE (due to POLE3-4 depletion) and decreased processivity of the remaining POLE (due to DSCC1 depletion), may therefore result in a lethal failure to complete DNA replication. Besides stabilizing Pole, POLE3-4 also exhibits histone cycling activity at DNA replication forks (Bellelli et al., 2018a). Interestingly, POLE3 Δ C, which lacks the H3-H4 interaction but retains the interaction with Pole components, displays defective PCNA unloading and RPA accumulation (Bellelli *et al.*, 2018a). However, POLE3 Δ C rescues ATR inhibitor sensitivity (Hustedt et al., 2019), suggesting that the H3-H4 binding is not required for mitigating DNA replication stress. Therefore, it remains to be determined whether the role of POLE3 as H3-H4 chaperone contributes to the observed synthetic lethality with DSCC1.”

2) Is the discovered function of MMS22L-TONSL shared also by Rtt101-Mms1 (DDB1-CUL4) shown to facilitate cohesion in yeast (Zhang et al, EMBO Reports, 2017)?

We address this point in an updated fourth paragraph of the Discussion:

“In yeast, Mms22 promotes Eco1-dependent Smc3 acetylation and SCC in collaboration with Rtt101-Mms1 (Zhang et al., 2017). This raises the question whether this role is conserved in human DDB1-Cul4. Indeed, DDB1 and Cul4A/B were reported to contribute to SCC in HEK293T cells via MMS22L-dependent ESCO2 recruitment (Sun et al., 2019). However, these genes were not synthetically lethal with DSCC1 in our CRISPR screen. This may be due

to poor efficacy of the used guide RNAs, and CUL4A and CUL4B may be partially redundant. Remarkably, DDB1-CUL4 has also been reported to induce ESCO2 degradation by interaction with ESCO2 via VRBP1, which is a substrate recognition component alternative to MMS22L (Minamino et al., 2018). This does not support a major role for CUL4-DDB1 in promoting SCC. Moreover, multiple proteomics studies did not detect an interaction of MMS22L with CUL4A/B or DDB1, but identify TONSL as the main MMS22L interactor in human cells (Duro et al., 2010; O'Connell et al., 2010; O'Donnell et al., 2010). Here we show that TONSL, which is absent in yeast, contributes to SCC establishment. This suggests potential differences between MMS22L function in yeast and human cells. MMS22L-TONSL can be recruited to chromatin by binding to H4K20me0, a mark of post-replicative chromatin, via a specific domain in TONSL (Saredi et al., 2016). Thereby, MMS22L-TONSL can discriminate replicated from unreplicated DNA (Saredi *et al.*, 2016), theoretically placing it in an ideal position to establish SCC by recruiting ESCO2 exclusively on newly replicated DNA. It will be interesting to assess a potential requirement for DDB1-CUL4 in this process.”

3) Mms1 and Mms22 were shown to stabilize the yeast replisome upon replication stress caused by HU treatment (Vaisica et al, MBC, 2011). Does MMS22LL functions jointly or in parallel with POLE3-POLE4 in ensuring replication?

Whereas POLE3 promotes normal fork progression, depletion of MMS22L appears not to lead to reduced fork speed (own observations and data by O'Donnell et al, 2010). Rather, MMS22L promotes the recovery from DNA replication stress (O'Donnell, 2010). This suggests that MMS22L and POLE3-4 have separate functions in DNA replication.

4) Although ESCO2 has a PIP in its N-terminus, the recruitment to chromatin in yeast can still happen in the absence of that domain (Moldovan et al, Mol Cell, 2006). Does MMS22-TONSL impair ESCO2 recruitment to chromatin, providing thus an alternative mechanism for ESCO2 recruitment and SCC establishment?

We agree with the reviewer that this is an important question. It is noticeable that ESCO2 contains three PCNA binding domains, as well as an MCM3 interaction domain and a MMS22L binding domain, that could contribute to its recruitment to chromatin. We now include a new Figure 4B and 4C to assess the ESCO2 recruitment in response to DSCC1/MMS22L depletion. DSCC1-ko cells show a reduction of ESCO2 and AcSMC3 on chromatin, likely due to a reduction of PCNA loading. MMS22L depletion also causes decreased ESCO2 levels on chromatin, which is further reduced by combined depletion of DSCC1 and MMS22L. This indicates that DSCC1 and MMS22L facilitate separate ESCO2 recruitment mechanisms: DSCC1 via PCNA loading and MMS22L via ESCO2's MMS22L interaction motif. This role of MMS22L/Mms22 in ESCO2 recruitment to chromatin is also supported by Zhang et al, 2017 and Sun et al, 2019.

Reviewer #2 (Comments to the Authors (Required)):

Sister chromatid cohesion (Cohesion) is crucial to ensure orderly segregation of chromosomes. Cohesion is established by co-entrapment of sister DNAs within cohesin rings during S-phase. The precise mechanism by which this is achieved is still unclear. A set of conserved proteins (but non-essential in yeast) associated with the replisome are important for cohesion establishment. They are thought to promote cohesion by two independent pathways that operate in parallel during S-phase. Analysis of cohesion of small circular mini-chromosomes in yeast suggests that Chl1 (DDX11), Ctf4 (AND1), Tof1 (TIMELESS) and

Csm3 (TIPIN) are important to build cohesion using DNA associated cohesin rings. The Ctf18-RFC (DSCC1-RFC) is important for loading soluble cohesin during S-phase to establish cohesion (Srinivasan et al eLife 2020).

In the present study, Janne van Schie et al characterize the DSCC1-RFC in human cells (Ctf18-RFC). They generate DSCC1 knockout cells (in a P53 deficient cell line) and find that this leads to cohesion defects, reduction in replication fork speed and accumulation of γ -H2AX foci. They then perform a CRISPR screen to look for genes that become essential in the absence of DSCC1 and identify several helicases (including DDX11), subunits of the polymerase epsilon complex and MMS22L. They find that loss of POLE3 in DSCC1-KO cells leads to additive defects specifically in DNA replication, while the loss of MMS21L leads to additive cohesion defects. Based on their observation that DDX11 is epistatic to MMS21L, they suggest that MMS21L and DSCC1 operate in parallel pathways for cohesion.

The observations reported here are interesting, they confirm the previously observed (in yeast and avian cells) genetic interaction between CTF18-RFC and DDX11. The present study also highlights the importance of MMS22L for cohesion in human cells (previously observed in yeast). The experiments are well executed, and the data are clear. The study reaffirms the evolutionary conservation of parallel pathways operating during S phase to establish cohesion and thus warrants publication. I have some comments:

We thank the reviewer for these kind words

1: The authors measure additive defects in DNA replication in the POLE3 DSSC1-KO double mutant cells, they conclude that the synthetic lethality of the DSCC1 cells upon POLE3 deletion is therefore specific to this additive defect DNA replication (perhaps reflected in the speed of replication). This might well be the case, there are reports in yeast cells that an interaction between CTF18-RFC and Pol epsilon is not necessary for cohesion (Grabarczyk et al 2018). However, comparing figure 1A with figure S3E it looks like there is a noticeable and perhaps significant increase in the cohesion defects in DSCC1-KO cells upon POLE3 deletion. Could the authors comment on this?

Although the reviewer is right that there appears to be a small difference, the extra effect of POLE3 in DSCC1-KO cells in fig S2E is marginal and seems insufficient to explain the synthetic lethality. Importantly, effects on SCC must be compared to controls within the same experiment, which makes a direct comparison with Fig 1A not relevant. In addition, POLE3 depletion in itself does not cause cohesion defects (Fig S2E) and the increase in mitosis following POLE3 depletion in DSCC1ko is small (Fig S2F). Therefore, replication defects are the most likely cause of the synthetic lethality between POLE3-4 and DSCC1.

2. Wapl depletion is shown to suppress the synthetic lethality of the CTF18 DDX11 double mutants in avian cells (Kawasumi et al 2021). If the authors wanted to make a strong case that the synthetic lethality of DSCC1 with POLE3 and DDX11/MMS21 was indeed because of specific defects in replication and cohesion respectively. One would expect Wapl depletion to suppress the synthetic lethality of DSCC1 with DDX11/MMS21 and not with POLE3. This is perhaps not within the remit of the current manuscript, in which case, the authors should be a bit cautious about assigning specificities for the synthetic interactions.

We agree that rescue by WAPL depletion would support that cohesion loss underlies the observed synthetic lethality of DSCC1 and MMS22L. We attempted this using WAPL crRNA and siRNA (see Figure 1 below). In this set-up, it turned out to be difficult to restore viability. In one of three experiments we observed a partial rescue in DDX11 depleted cells and also a

small effect in MMS22L depleted cells, possibly related to a different timing (Fig. 1C). Our observations are indeed different from those reported from chicken DT40 cells, in which auxin-mediated WAPL depletion suppressed lethality in CTF18-DDX11 double mutants (Kawasumi et al., 2021). Since prolonged WAPL depletion can also impair growth rate, it may be difficult to balance the effects of different gene depletions. Moreover, WAPL depletion could suppress lethality neither in CTF18-Chl1 double mutant in yeast (Kawasumi et al., 2021), nor of ESCO1-ESCO2 depleted vertebrate cells (Kawasumi et al., 2017). As the reviewer indicates, solving the exact degree and timing of toxicity of different forms of impaired SCC establishment is not within the scope of this manuscript, although admittedly it is an important question for the future. We agree that we cannot exclude that mechanisms other than enhanced cohesion loss contribute to the observed synthetic lethality of DSCC1 and MMS22L, which we now indicate in the manuscript (p.5, line 6-9).

Figure 1: Impact of WAPL depletion on viability in DSCC1ko cells depleted of DDX11, POLE3 or MMS22L.

A, RPE1-iCas9-DSCC1ko cells were treated with 200 ng/mL doxycycline to induce Cas9 expression and transfected with crOR10A7 (control), crDDX11, crPOLE3 or crMMS22L; in all cases combined with either crOR10A7 or crWAPL. After three days, 1000 cells per well were seeded in a 96-wells plate and viability was measured after another six days using Cell-Titer Blue.

B, Cells were treated with doxycycline and transfected with indicated crRNAs. After three days, 1000 cells per well were transfected with non-targeting siRNA or siWAPL in a 96-wells plate. After another five days, viability was measured.

C, Cells were treated and transfected as in A. After two days, 1000 cells per well were seeded in a 96-wells plate and viability was measured after another five days.

3. In Discussion the authors state: Furthermore, since it is conceivable that all cohesin needs to be acetylated in order to contribute to SCC, it is unclear why DSCC1-RFC would specifically contribute to the de novo loading pathway (Srinivasan et al., 2020).

There is a flaw in their logic here. Eco1/ESCO1 abbreviated for establishment of cohesion is a misnomer. It is clear in almost every living system tested thus far that Eco1 is not fundamental for establishment of cohesion in the absence of Wapl mediated release. The analysis of ctf18 mutants in yeast that were defective in Wapl mediated release (and therefore emancipated from Eco1) revealed that they were defective in de novo generation of cohesion in S-phase. The authors observe that levels of Smc3, acetylated Smc3 as well as Sororin on chromatin is lower in DSCC1-KO cells and this is exacerbated by deletion of DDX11 or MMS22L. These observations are consistent with a defect in generation of cohesion in the absence of DSCC1-RFC.

We agree with this description, but we feel that it does not challenge our point of view (note that we use WAPL proficient cells). To better clarify our logic, we have now considerably altered this paragraph in the Discussion (third paragraph)

“DSSC1 functions in the *de novo* loading pathway to establish SCC (Kawasumi et al., 2021; Srinivasan et al., 2020), in line with the synthetic lethality with DDX11 observed in this study. A recent preprint of the Branzei lab shows that PCNA can directly interact with the cohesin loader in yeast and human cells, thus providing a mechanistic explanation for the role of the PCNA loader DSSC1-RFC in SCC (Psakhye et al., 2022). In addition, PCNA is known to recruit ESCO2 and thereby promotes SMC3 acetylation (Bender et al., 2020; Liu et al., 2020; Moldovan et al., 2006). Indeed we find partially impaired ESCO2 recruitment and SMC3 acetylation in DSSC1-KO cells. Interestingly, we also present evidence that MMS22L promotes a separate ESCO2 recruitment pathway: its depletion results in further reduced ESCO2 recruitment, additive cohesion defects and lethality in DSSC1-KO cells, but is epistatic in viability with DDX11. These genetic interactions place MMS22L in the cohesin conversion pathway. Similarly, the function of yeast Mms22 in cohesion is epistatic with the cohesin conversion component Ctf4 (Zhang *et al.*, 2017), in line with a physical interaction of Mms22 and Ctf4 (Gambus et al., 2009; Mimura et al., 2010). Since both PCNA and MMS22L can recruit ESCO2 to the replisome (Bender *et al.*, 2020; Sun *et al.*, 2019; Zhang *et al.*, 2017), we propose that different SCC establishment pathways in part rely on different mechanisms to recruit ESCO2 to the replisome (Fig 4F). It will be interesting to determine if ESCO2, recruited by either PCNA or MMS22L, exhibits different preferences for cohesin arising from different SCC establishment pathways.”

Reviewer #3 (Comments to the Authors (Required)):

The authors investigate DSSC1-RFC, the leading strand alternative PCNA clamp loader. DSSC1-RFC facilitates DNA replication, repair and sister chromatid cohesion (SSC). Here, they perform a genome-wide Crispr screen to find mutated human genes that have negative synthetic interactions with DSSC1-KO. Previous work identified two pathways involving replisome associated proteins in establishment of SSC, one defined by DSSC1 and the other by DDX11. In a model proposed by Nasmyth and coworkers, DSSC1 generates cohesion with cohesin loaded at the replication fork whereas DDX11 generates cohesion with cohesin loaded on unreplicated DNA. As shown here and elsewhere, synthetic lethality occurs when both DSSC1 and DDX11 are mutated. The author's Crispr screen identified many genes involved in DNA replication and cell cycle progression but they follow up with just two. The first is MMS22L, which together with TONSL forms a protein complex at the replication fork. As shown here and in yeast, MMS22L participates in SSC. Since the authors show that MMS22L and DSSC1 contribute to SCC in additive ways, they conclude that they act in different but parallel pathways. They infer that MMS22L-TONSL contributes via the pathway defined by DDX11, which is supported by earlier prior work in yeast. Their work builds on work from Nasmyth and Zou labs by showing that human MMS22L participates in a different cohesion establishment pathway (presumably the DDX1 pathway) than the pathway defined by DSSC1. Putting all this together in humans by an unbiased screen is a timely and worthy advance.

The second gene characterized is POLE3, which together with POLE4, generates a protein complex that, like DSSC1, associates with the leading strand polymerase. They show that POLE3 and DSSC1 contribute to replication in additive ways, suggesting that they act in different but parallel pathways. Only loss of DSSC1 causes H2AX foci, a marker of DNA damage. POLE3 mutation does not create cohesion defects on its own and does not exacerbate defects in DSSC1-KO. Thus, the synthetic relationship between the two is not likely related to cohesion.

Major issues:

1) In figure 4C, the authors claim that there is an additive effect on levels of chromatin-bound cohesin, Smc3 acetylation and sororin when MMS22L and DSSC1 are co-depleted. That effect is imperceptible by eye, particularly when compounded by slight differences in loading (the H3 control), which are noticeable by eye. Without quantitation, I am not convinced. The weakness of this panel maybe reflected by the fact that the authors don't mention it in the Discussion. This should be remedied with quantitation or removed.

We now include a new Figure 4B and 4C to assess the ESCO2 recruitment to chromatin and AcSMC3 in response to DSSC1/MMS22L depletion. DSSC1-ko cells show a clear reduction of ESCO2 and AcSMC3. MMS22L depletion also causes decreased ESCO2 levels and has a small effect on AcSMC3. Combined depletion of DSSC1 and MMS22L further reduced ESCO2 and AcSMC3 on chromatin. This suggests that DSSC1 and MMS22L facilitate separate ESCO2 recruitment mechanisms: DSSC1 via PCNA loading and MMS22L via ESCO2's MMS22L interaction motif. This is included in the Discussion (third paragraph).

2) In figure 4C, the band labelled "Total Smc3" should just say Smc3 since it is only measuring the Smc3 in the chromatin pellet. This experiment should also include a control that either shows that the Smc3 level in whole cell lysate does not change or that the level of Smc3 in the soluble fraction increases.

We agree with the reviewer and included these changes in the updated Figure 4C.

3) In figure 4A, the authors show that MMS22L and DDX11 depletions both cause additive effects on cohesion when either is combined with DSSC1-KO. This suggests that both operate in pathways parallel to DSSC1. Does this mean they operate in the same parallel pathway? It should be possible to test by co-depletion of MMS22L and DDX11. Otherwise, the authors should reiterate that MMS22L and DDX11 are placed in the same pathway by the yeast data of Zou et al.

Figure 4D shows that MMS22L is epistatic with DDX11 on viability, whereas it is synthetically lethal with DSSC1. Considering the severe effect on cohesion following single depletion of MMS22L or DDX11, an additive effect on cohesion would most likely have an impact on viability as well. The observed epistasis therefore hints that MMS22L and DDX11 operate in the same cohesion establishment pathway. Mms22 was reported to interact physically with Ctf4, another component of the cohesin conversion pathway (Gambus et al, 2009 and Mimura et al, 2010), and Mms22 function in cohesion is epistatic with Ctf4 (Zhang et al, 2017). We updated the third paragraph of the discussion where we address this issue.

3) Discussion - Zou and coworkers showed that MMS22L and its partners (DDB and CUL4) are required for proper cohesion in yeast. Notably, these MMS22L partner proteins were not picked up in the Crispr Screen. This is troubling because the authors argue that synthetic lethality arises from compounding cohesion loss by depletion of both MMS22L and DSSC1. This should either be tested directly by depleting DDB and CUL4 in the DSSC1-KO, or by finding a better explanation. In the second to last paragraph, the argument that MMS22L must have "shared and separate" roles in ESCO2 regulation is irrelevant if cohesion is the only role that matters for synthetic lethality.

We assume that the reviewer refers to two papers from the Lou laboratory (Zhang *et al.*, 2017 and Sun *et al.*, 2019). We agree that the putative involvement of DDB1-Cul4 required more extensive discussion, which we now include in the fourth paragraph of the Discussion:

“In yeast, Mms22 promotes Eco1-dependent Smc3 acetylation and SCC in collaboration with Rtt101-Mms1 (Zhang *et al.*, 2017). This raises the question whether this role is conserved in human DDB1-Cul4. Indeed, DDB1 and Cul4A/B were reported to contribute to SCC in HEK293T cells via MMS22L-dependent ESCO2 recruitment (Sun *et al.*, 2019). However, these genes were not synthetically lethal with DSCC1 in our CRISPR screen. This may be due to poor efficacy of the used guide RNAs, and CUL4A and CUL4B may be partially redundant. Remarkably, DDB1-CUL4 has also been reported to induce ESCO2 degradation by interaction with ESCO2 via VRBP1, which is a substrate recognition component alternative to MMS22L (Minamino *et al.*, 2018). This does not support a major role for CUL4-DDB1 in promoting SCC. Moreover, multiple proteomics studies did not detect an interaction of MMS22L with CUL4A/B or DDB1, but identify TONSL as the main MMS22L interactor in human cells (Duro *et al.*, 2010; O'Connell *et al.*, 2010; O'Donnell *et al.*, 2010). Here we show that TONSL, which is absent in yeast, contributes to SCC establishment. This suggests potential differences between MMS22L function in yeast and human cells. MMS22L-TONSL can be recruited to chromatin by binding to H4K20me0, a mark of post-replicative chromatin, via a specific domain in TONSL (Saredi *et al.*, 2016). Thereby, MMS22L-TONSL can discriminate replicated from unreplicated DNA (Saredi *et al.*, 2016), theoretically placing it in an ideal position to establish SCC by recruiting ESCO2 exclusively on newly replicated DNA. It will be interesting to assess a potential requirement for DDB1-CUL4 in this process.”

Minor issues:

1) Page 4, first paragraph - Reference to figure 3F is wrong.

We thank the reviewer for noticing this mistake.

2) Top sentence of figure 6, "but is epistatic with DDX11." - Clarify that epistasis regards viability (fig 4D-E), not cohesion (fig 4A).

We modified this.

4) Middle of page 3 - "This may relate to the role of DSCC1-RFC in de novo cohesin loading" should read "de novo cohesin loading pathway" since the authors acknowledge that the mechanistic bases for the two pathways have not been worked out, as they properly state in the last paragraph of the Discussion.

We modified this.

5) On page 5 - "cohesion defects caused by DSCC1-RFC loss cannot be rescued by ESCO1 or ESCO2 overexpression (Kawasumi 2021)." In my view this is a tricky citation since it is a negative result, and there are many ways that an overexpression experiment can fail.

We now include an updated third paragraph of the Discussion, in which we omitted this reference.

6) The figure legends are skeletal, thereby forcing the reader to look in other places to find out what was done exactly.

We updated the figure legends.

References

- Bellelli, R., Belan, O., Pye, V.E., Clement, C., Maslen, S.L., Skehel, J.M., Cherepanov, P., Almouzni, G., and Boulton, S.J. (2018a). POLE3-POLE4 Is a Histone H3-H4 Chaperone that Maintains Chromatin Integrity during DNA Replication. *Mol Cell* *72*, 112-126 e115. 10.1016/j.molcel.2018.08.043.
- Bellelli, R., Borel, V., Logan, C., Svendsen, J., Cox, D.E., Nye, E., Metcalfe, K., O'Connell, S.M., Stamp, G., Flynn, H.R., et al. (2018b). Polepsilon Instability Drives Replication Stress, Abnormal Development, and Tumorigenesis. *Mol Cell* *70*, 707-721 e707. 10.1016/j.molcel.2018.04.008.
- Bender, D., Da Silva, E.M.L., Chen, J., Poss, A., Gawey, L., Rulon, Z., and Rankin, S. (2020). Multivalent interaction of ESCO2 with the replication machinery is required for sister chromatid cohesion in vertebrates. *Proceedings of the National Academy of Sciences* *117*, 1081-1089. 10.1073/pnas.1911936117.
- Duro, E., Lundin, C., Ask, K., Sanchez-Pulido, L., MacArtney, T.J., Toth, R., Ponting, C.P., Groth, A., Helleday, T., and Rouse, J. (2010). Identification of the MMS22L-TONSL complex that promotes homologous recombination. *Mol Cell* *40*, 632-644. 10.1016/j.molcel.2010.10.023.
- Gambus, A., van Deursen, F., Polychronopoulos, D., Foltman, M., Jones, R.C., Edmondson, R.D., Calzada, A., and Labib, K. (2009). A key role for Ctf4 in coupling the MCM2-7 helicase to DNA polymerase alpha within the eukaryotic replisome. *EMBO J* *28*, 2992-3004. 10.1038/emboj.2009.226.
- Garcia-Rodriguez, L.J., De Piccoli, G., Marchesi, V., Jones, R.C., Edmondson, R.D., and Labib, K. (2015). A conserved Pol binding module in Ctf18-RFC is required for S-phase checkpoint activation downstream of Mec1. *Nucleic Acids Res* *43*, 8830-8838. 10.1093/nar/gkv799.
- Grabarczyk, D.B., Silkenat, S., and Kisker, C. (2018). Structural Basis for the Recruitment of Ctf18-RFC to the Replisome. *Structure* *26*, 137-144.e133. 10.1016/J.STR.2017.11.004.
- Hustedt, N., Alvarez-Quilon, A., McEwan, A., Yuan, J.Y., Cho, T., Koob, L., Hart, T., and Durocher, D. (2019). A consensus set of genetic vulnerabilities to ATR inhibition. *Open Biol* *9*, 190156. 10.1098/rsob.190156.
- Kawasumi, R., Abe, T., Psakhye, I., Miyata, K., Hirota, K., and Branzei, D. (2021). Vertebrate CTF18 and DDX11 essential function in cohesion is bypassed by preventing WAPL-mediated cohesin release. *Genes Dev.* 10.1101/gad.348581.121.
- Liu, H.W., Bouchoux, C., Panarotto, M., Kakui, Y., Patel, H., and Uhlmann, F. (2020). Division of Labor between PCNA Loaders in DNA Replication and Sister Chromatid Cohesion Establishment. *Molecular Cell* *78*, 725-738.e724. 10.1016/j.molcel.2020.03.017.
- Mimura, S., Yamaguchi, T., Ishii, S., Noro, E., Katsura, T., Obuse, C., and Kamura, T. (2010). Cul8/Rtt101 forms a variety of protein complexes that regulate DNA damage response and transcriptional silencing. *J Biol Chem* *285*, 9858-9867. 10.1074/jbc.M109.082107.
- Minamino, M., Tei, S., Negishi, L., Kanemaki, M.T., Yoshimura, A., Sutani, T., Bando, M., and Shirahige, K. (2018). Temporal Regulation of ESCO2 Degradation by the MCM

Complex, the CUL4-DDB1-VPRBP Complex, and the Anaphase-Promoting Complex. *Current Biology* 28, 2665-2672.e2665. 10.1016/J.CUB.2018.06.037.

Moldovan, G.L., Pfander, B., and Jentsch, S. (2006). PCNA Controls Establishment of Sister Chromatid Cohesion during S Phase. *Molecular Cell* 23, 723-732. 10.1016/j.molcel.2006.07.007.

Murakami, T., Takano, R., Takeo, S., Taniguchi, R., Ogawa, K., Ohashi, E., and Tsurimoto, T. (2010). Stable interaction between the human proliferating cell nuclear antigen loader complex Ctf18-replication factor C (RFC) and DNA polymerase {epsilon} is mediated by the cohesion-specific subunits, Ctf18, Dcc1, and Ctf8. *J Biol Chem* 285, 34608-34615. 10.1074/jbc.M110.166710.

O'Connell, B.C., Adamson, B., Lydeard, J.R., Sowa, M.E., Ciccio, A., Bredemeyer, A.L., Schlabach, M., Gygi, S.P., Elledge, S.J., and Harper, J.W. (2010). A genome-wide camptothecin sensitivity screen identifies a mammalian MMS22L-NFKBIL2 complex required for genomic stability. *Mol Cell* 40, 645-657. 10.1016/j.molcel.2010.10.022.

O'Donnell, L., Panier, S., Wildenhain, J., Tkach, J.M., Al-Hakim, A., Landry, M.C., Escobedo-Diaz, C., Szilard, R.K., Young, J.T., Munro, M., et al. (2010). The MMS22L-TONSL complex mediates recovery from replication stress and homologous recombination. *Mol Cell* 40, 619-631. 10.1016/j.molcel.2010.10.024.

Psakhye, I., Kawasumi, R., Abe, T., Hirota, K., and Branzei, D. (2022). PCNA recruits cohesin loader Scc2/NIPBL to ensure sister chromatid cohesion. *BioRxiv*. <https://doi.org/10.1101/2022.09.16.508217>.

Saredi, G., Huang, H., Hammond, C.M., Alabert, C., Bekker-Jensen, S., Forne, I., Reveron-Gomez, N., Foster, B.M., Mlejnkova, L., Bartke, T., et al. (2016). H4K20me0 marks post-replicative chromatin and recruits the TONSL-MMS22L DNA repair complex. *Nature* 534, 714-718. 10.1038/nature18312.

Srinivasan, M., Fumasoni, M., Petela, N.J., Murray, A., and Nasmyth, K.A. (2020). Cohesion is established during dna replication utilising chromosome associated cohesin rings as well as those loaded de novo onto nascent dnas. *eLife* 9, 1-27. 10.7554/eLife.56611.

Stokes, K., Winczura, A., Song, B., Piccoli, G., and Grabarczyk, D.B. (2020). Ctf18-RFC and DNA Pol form a stable leading strand polymerase/clamp loader complex required for normal and perturbed DNA replication. *Nucleic Acids Res* 48, 8128-8145. 10.1093/nar/gkaa541.

Sun, H., Zhang, J., Xin, S., Jiang, M., Zhang, J., Li, Z.I., Cao, Q.I., and Lou, H.I. (2019). Cul4-Ddb1 ubiquitin ligases facilitate DNA replication-coupled sister chromatid cohesion through regulation of cohesin acetyltransferase Esco2. 10.1371/journal.pgen.1007685.

Zhang, J., Shi, D., Li, X., Ding, L., Tang, J., Liu, C., Shirahige, K., Cao, Q., and Lou, H. (2017). Rtt101-Mms1-Mms22 coordinates replication-coupled sister chromatid cohesion and nucleosome assembly. *EMBO reports* 18, 1294-1305. 10.15252/EMBR.201643807.

November 28, 2022

RE: Life Science Alliance Manuscript #LSA-2022-01596R

Dr. Job de Lange
Amsterdam University Medical Centers
Human Genetics
de Boelelaan 1118
Amsterdam 1081 HV
Netherlands

Dear Dr. de Lange,

Thank you for submitting your revised manuscript entitled "MMS22L-TONSL functions in sister chromatid cohesion in a pathway parallel to DSCC1-RFC". We would be happy to publish your paper in Life Science Alliance pending final revisions necessary to meet our formatting guidelines.

- please add the Twitter handle of your host institute/organization as well as your own or/and one of the authors in our system
- please add the author contributions and a conflict of interest statement to the main manuscript text
- please use the [10 author names, et al.] format in your references (i.e. limit the author names to the first 10)

A. FINAL FILES:

B. MANUSCRIPT ORGANIZATION AND FORMATTING:

**Submission of a paper that does not conform to Life Science Alliance guidelines will delay the acceptance of your

manuscript.**

The license to publish form must be signed before your manuscript can be sent to production. A link to the electronic license to publish form will be sent to the corresponding author only. Please take a moment to check your funder requirements.

Sincerely,

Reviewer #1 (Comments to the Authors (Required)):

This study provides information on the roles of DSCC1 and MMS21L in cohesion versus other characterized pathways. The paper has been improved during revision and warrants publication in Life Science Alliance.

Reviewer #2 (Comments to the Authors (Required)):

Van Schie et al have satisfactorily addressed the referees comments. The revised manuscript highlights the role of DScC1-RFC and MMS21L in recruitment of Esco2 to the replication forks in order to acetylate cohesin and stabilise cohesion. I stand by my initial assessment: The experiments are well executed, and the data are clear. The study reaffirms the evolutionary conservation of parallel pathways operating during S phase to establish cohesion and warrants publication.

Reviewer #3 (Comments to the Authors (Required)):

The revised manuscript by Schie and de Lange dealt adequately with my concerns with the first submission. This work represents a timely and worthy advance. I do not seek further revision.

November 30, 2022

RE: Life Science Alliance Manuscript #LSA-2022-01596RR

Dr. Job de Lange
Amsterdam University Medical Centers
Human Genetics
de Boelelaan 1118
Amsterdam 1081 HV
Netherlands

Dear Dr. de Lange,

Thank you for submitting your Research Article entitled "MMS22L-TONSL functions in sister chromatid cohesion in a pathway parallel to DSCC1-RFC". It is a pleasure to let you know that your manuscript is now accepted for publication in Life Science Alliance. Congratulations on this interesting work.

DISTRIBUTION OF MATERIALS:

Again, congratulations on a very nice paper. I hope you found the review process to be constructive and are pleased with how the manuscript was handled editorially. We look forward to future exciting submissions from your lab.

Sincerely,
